# STIM1 Deficiency Leads to Specific Down-Regulation of ITPR3 in SH-SY5Y Cells

**DOI:** 10.3390/ijms21186598

**Published:** 2020-09-09

**Authors:** Carlos Pascual-Caro, Yolanda Orantos-Aguilera, Irene Sanchez-Lopez, Jaime de Juan-Sanz, Jan B. Parys, Estela Area-Gomez, Eulalia Pozo-Guisado, Francisco Javier Martin-Romero

**Affiliations:** 1Department of Biochemistry and Molecular Biology, School of Life Sciences and Institute of Molecular Pathology and Biomarkers, University of Extremadura, 06006 Badajoz, Spain; carlospc@unex.es (C.P.-C.); yorantosa@unex.es (Y.O.-A.); isanchezlo@unex.es (I.S.-L.); 2Sorbonne Universités and Institut du Cerveau et de la Moelle Epinière (ICM) - Hôpital Pitié-Salpêtrière, Inserm, CNRS, 75013 Paris, France; jaime.dejuansanz@icm-institute.org; 3Department of Cellular and Molecular Medicine, Leuven Kanker Instituut, KU Leuven, B-3000 Leuven, Belgium; jan.parys@kuleuven.be; 4Department of Neurology, Columbia University Medical Center, New York, NY 10032-3748, USA; eag2118@cumc.columbia.edu; 5Department of Cell Biology, School of Medicine and Institute of Molecular Pathology and Biomarkers, University of Extremadura, 06006 Badajoz, Spain; epozo@unex.es

**Keywords:** calcium, endoplasmic reticulum, IP3 receptor, mitochondria, neurodegeneration, STIM1

## Abstract

STIM1 is an endoplasmic reticulum (ER) protein that modulates the activity of a number of Ca^2+^ transport systems. By direct physical interaction with ORAI1, a plasma membrane Ca^2+^ channel, STIM1 activates the *I_CRAC_* current, whereas the binding with the voltage-operated Ca^2+^ channel Ca_V_1.2 inhibits the current through this latter channel. In this way, STIM1 is a key regulator of Ca^2+^ signaling in excitable and non-excitable cells, and altered STIM1 levels have been reported to underlie several pathologies, including immunodeficiency, neurodegenerative diseases, and cancer. In both sporadic and familial Alzheimer’s disease, a decrease of STIM1 protein levels accounts for the alteration of Ca^2+^ handling that compromises neuronal cell viability. Using SH-SY5Y cells edited by CRISPR/Cas9 to knockout *STIM1* gene expression, this work evaluated the molecular mechanisms underlying the cell death triggered by the deficiency of STIM1, demonstrating that STIM1 is a positive regulator of *ITPR3* gene expression. ITPR3 (or IP3R3) is a Ca^2+^ channel enriched at ER-mitochondria contact sites where it provides Ca^2+^ for transport into the mitochondria. Thus, STIM1 deficiency leads to a strong reduction of *ITPR3* transcript and ITPR3 protein levels, a consequent decrease of the mitochondria free Ca^2+^ concentration ([Ca^2+^]_mit_), reduction of mitochondrial oxygen consumption rate, and decrease in ATP synthesis rate. All these values were normalized by ectopic expression of ITPR3 in STIM1-KO cells, providing strong evidence for a new mode of regulation of [Ca^2+^]_mit_ mediated by the STIM1-ITPR3 axis.

## 1. Introduction

Calcium (Ca^2+^) transport through biological membranes is essential to control a wide range of intracellular signaling pathways. This transport is tightly regulated to maintain a cytosolic free Ca^2+^ concentration ([Ca^2+^]_i_) close to 100 nM in resting conditions, and the transient increase over this basal level acts as an effector for many stimuli. There are multiple signaling pathways that become deregulated during cytosolic Ca^2+^ dyshomeostasis, which is the basis for the proposal of the Ca^2+^ hypothesis of neurodegenerative diseases that suggests a direct causal link between altered Ca^2+^ signaling and the onset of Alzheimer’s [1,2,3] and Parkinson’s diseases [4,5].

Amongst the many regulators of intracellular Ca^2+^ concentration, STIM1 stands out as a master regulator of Ca^2+^ channels. STIM1 is a transmembrane protein located in the endoplasmic reticulum (ER) that senses the intraluminal Ca^2+^ concentration through an EF-hand domain close to the N-terminus of the protein [6]. The inositol 1,4,5-trisphosphate (IP3)-mediated Ca^2+^ release from the ER leads to the transient lowering of the ER Ca^2+^ concentration and subsequent activation of STIM1. This activation involves the unfolding of the STIM1 cytoplasmic domain in an extended conformation, that enables this protein to bind plasma membrane Ca^2+^ channels [7]. Therefore, STIM1 regulates Ca^2+^ transport through the plasma membrane in an intracellular Ca^2+^ store-dependent manner, an action that has been termed store-operated Ca^2+^ entry (SOCE). The molecular binding of STIM1 to the plasma membrane Ca^2+^ channel ORAI1 has been extensively studied (reviewed in References [8,9]). However, STIM1 is also involved in both, the negative regulation of Ca_V_1.2, an L-type voltage-operated Ca^2+^ channel [10,11,12,13], and of Ca_V_3.1, a T-type voltage-operated Ca^2+^ channel [14], by direct interaction with the channel itself. Other channels and receptors are also regulated by STIM1, such as the -amino-3-hydroxy-5-methyl-4-isoxazole propionic acid (AMPA) receptor subunit GluR1 or GluA1 (the product of the gene *GRIA1*) [15], or members of the transient receptor potential canonical channel (TRPC) family of channels (reviewed in Reference [16]), confirming the role of STIM1 in the control of a wide range of receptors and channels.

Given this range of channels modulated by STIM1, it is not surprising that there is growing evidence supporting a key role for STIM1 in neuronal function, synaptic strength, and memory formation [17,18,19]. Conversely, alterations in STIM1 expression and/or activity lead to neuronal dysfunction. In this regard, it is known that presenilin 1 (PSEN1), a catalytic subunit of the γ-secretase complex, co-precipitates with STIM1, and that the treatment of SH-SY5Y cells with N-[N-(3,5-difluorophenacetyl)-L-alanyl]-S-phenylglycine t-butyl ester (DAPT), a known inhibitor of PSEN1, increases the levels of SOCE. This set of results suggests that STIM1 is targeted by the γ-secretase complex by cleavage of the transmembrane domain [20].

On the other hand, most PSEN1 mutations are loss-of-function mutations, but there are mutations in PSEN1 associated with familial Alzheimer’s disease (FAD) that can be considered as gain-of-function mutations. These are M146L, V97L, A136G, and A246E [21,22]. These mutations lead to an alteration of the amyloid precursor protein (APP) processing, increasing the ratio Aβ42/Aβ40, so it is generally assumed that these mutations seem to represent a gain-of-function mode. In this regard, fibroblasts from patients with FAD-related PSEN1 variants, such as M146L, show lowered levels of total STIM1, SOCE inhibition, and decrease of STIM1 clustering rates [20]. Similarly, overexpression of PSEN1 with FAD-associated mutations led to an attenuation of SOCE in HEK293 cells [23].

These results suggest that aberrant STIM1 levels could underlie the pathogenesis in FAD. In this regard, we recently reported that there is a significant reduction of STIM1 levels in human brain tissues (medium frontal gyrus) from patients diagnosed with sporadic (non-familial) Alzheimer’s disease (SAD) [24]. The real impact of the loss of STIM1 at the cellular level was studied by knocking-out *STIM1* gene in the neuroblastoma cell line SH-SY5Y by genome editing with CRISPR/Cas9. Mitochondria from STIM1-knockout (KO) cells did show lower levels of free Ca^2+^ concentration ([Ca^2+^]_mit_). While it is well known that this [Ca^2+^]_mit_ is essential to control mitochondria functionality [25], it remains unknown why mitochondria in STIM1-deficient cells do not reach the [Ca^2+^]_mit_ observed in wild-type cells.

Interestingly, mitochondrial dysfunction is a key feature in Alzheimer’s disease, as mitochondria from Alzheimer’s disease (AD) patients show decreased uptake of Ca^2+^ [26]. In this report, we studied the role of STIM1 in the normalization of [Ca^2+^]_mit_ by analyzing the gene expression of some of the most important Ca^2+^ transport systems. Our data revealed that the expression of IP3 receptor type 3 (IP3R3 or ITPR3) is significantly lowered in STIM1-deficient cells. This alteration in ITPR3 expression was found to be responsible for the low [Ca^2+^]_mit_ since the ectopic overexpression of ITPR3 in STIM1-KO cells normalized [Ca^2+^]_mit_, as well as basal oxygen consumption rate, mitochondrial coupling efficiency, ATP production rate, and cell viability.

## 2. Results

### 2.1. Expression of ITPR3 Is Modulated by STIM1 Protein Levels

As stated above, [Ca^2+^]_mit_ is critical to support mitochondrial function. We recently reported that human brain samples from SAD patients present significant decreases in the levels of STIM1 [24]. Using SH-SY5Y cells as an in vitro model system, we showed that this decline in STIM1 leads to reductions in [Ca^2+^]_mit_ and the inhibition of mitochondrial complex I (NADH-coenzyme Q oxidoreductase) activity. However, the molecular mechanism underlying these alterations is not clear.

Impairments in the transport of Ca^2+^ into mitochondria could be caused by alterations in the levels or the activity of a wide range of Ca^2+^ transporters and signaling molecules. To find the regulators responsible for this impairment, we monitored the expression of different transcripts included in the Human Intracellular Calcium Signaling TaqMan Gene Expression Assay (Appendix A). The results revealed a significant drop in the expression of the *ITPR3* gene in STIM1-KO SH-SY5Y cells compared to controls (Appendix A and Figure 1a).

It is well-known that the efficiency of Ca^2+^ shuttling between ER and mitochondria relies on the activity of inositol 1,4,5-trisphosphate receptors (IP3Rs or ITPRs) [27,28,29]. We, therefore, studied in depth the possibility that the low [Ca^2+^]_mit_ found in STIM1-KO cells was caused by reductions in the expression levels of ITPR3. To confirm the TaqMan results, we analyzed mRNA levels for *ITPR1*, *ITPR2*, and *ITPR3* genes by quantitative RT-PCR in non-differentiated SH-SY5Y cells. In agreement with the results shown above, we found ~90% reduction in *ITPR3* gene transcripts, whereas the products of *ITPR1* and *ITPR2* did not change significantly (Figure 1b). Remarkably, overexpression of STIM1 in SH-SY5Y cells (Figure A1, top panel) significantly increased the level of *ITPR3* transcripts, without altering the level of *ITPR1-2* (Figure A1, bottom panel), which suggests a positive correlation between the expression of STIM1 and the levels of ITPR3. Immunoblot analyses confirmed that there was a significant decrease of ITPR3 protein in STIM1-KO cells, which was <30% of that found in the parental wild-type cell line (Figure 1c). A similar result was observed in differentiated SH-SY5Y cells (Figure A2). On the contrary, the level of ITPR1 and ITPR2 proteins were not significantly altered by the absence of STIM1 (Figure 1d,e), as it was shown for *ITPR1/2* transcripts (Figure 1b).

The decline of ITPR3 expression could have impact on the overall regulation of Ca^2+^ signaling in the cell. However, the comparison of ITPR3 expression between SH-SY5Y and U2OS cells, where ITPR3 is the most abundant isoform, suggests that this specific variant is weakly expressed in SH-SY5Y cells (Figure 1f, left panel), as shown before [30]. In addition, the use of an antibody that recognizes the three isoforms did not show any difference between wild-type SH-SY5Y and STIM1-KO SH-SY5Y cells (Figure 1f, right panel), further confirming that the expression of ITPR3 represents a minor contribution to the total level of ITPRs in SH-SY5Y cells.

The importance of ITPR3 relies on the fact that this specific isoform was found to be enriched at the interactions between ER and mitochondria (mitochondria-associated ER membranes (MAMs)) in different cell lines [27,28,29], where it regulates the transfer of Ca^2+^ between these organelles. This specific localization of ITPR3 in MAMs was also found in SH-SY5Y cells in this work (Figure A3).

### 2.2. Functional Consequences of the ITPR3 Downregulation

To study the functional consequences of the ITPR3 downregulation in STIM1-deficient cells, we assessed the release of Ca^2+^ from the ER in fura-2-loaded cells stimulated with 100 μM carbachol (CCh) in Ca^2+^-free medium. This stimulation, which activates the IP3 pathway, revealed a partial decrease in the amplitude of the Ca^2+^ release from the ER in response to a 30 s pulse of CCh (Figure 2a). Significantly, we were able to recapitulate these results in differentiated cells (Figure 2b).

Alterations in the gradient of Ca^2+^ between the intraluminal space of the ER and the cytosol could also result in reductions in the release of Ca^2+^ from the ER. To test for this possibility, we measured the intraluminal free Ca^2+^ concentration in the ER ([Ca^2+^]_ER_) in resting conditions by transfecting the genetically-encoded Ca^2+^ sensor ER-GCaMP6-210 [31]. Our results showed there to be a statistically significant but slight decrease of the [Ca^2+^]_ER_ in STIM1-deficient cells (from 246 ± 10.8 μM in wild-type cells to 202 ± 11.7 μM in STIM1-KO cells) (Figure 3a). However, it is important to note here that the measured cytosolic Ca^2+^ concentration ([Ca^2+^]_i_) in wild-type cells was greater than that found in STIM1-KO cells (77.4 ± 10 nM vs 44 ± 4.4 nM; see data in Reference [24]), so that the gradient of the [Ca^2+^] across the ER membrane differed little between wild-type and STIM1-KO cells and cannot explain the diminished Ca^2+^ release in STIM1-deficient cells in response to CCh.

Therefore, we analyzed the speed of ER emptying by measuring the time constant tau (τ) of the kinetics of [Ca^2+^]_ER_ in response to CCh. This ER emptying follows a first-order exponential decay (Figure 3b), with the τ-value of this kinetics being significantly increased by the reduction in STIM1, rising from 10.5 ± 0.48 s (for wild-type cells) to 14.29 ± 0.94 s (for STIM1-KO cells, and n = 26 regions of interest, ROIs, in each experimental condition). In agreement with the reductions in ITPR3 levels in STIM1-KO cells shown in Figure 1, these data support a slower Ca^2+^ release from the ER in STIM1-KO cells. These results, which illustrate a deficient fast release of Ca^2+^ from the ER in STIM1-deficient cells, are of significant importance considering the localization of ITPR3, which were found highly enriched in MAMs (Figure A3).

### 2.3. Overexpression of ITPR3 Restores [Ca^2+^]_mit_ in STIM1-KO Cells

To investigate whether deficiency in STIM1 and the consequent lowering of ITPR3 levels modified the regulation of the mitochondrial Ca^2+^ buffering capacity, we measured the kinetics of the mitochondrial Ca^2+^ concentration in SH-SY5Y cells, expressing the Ca^2+^ sensor mito^4×^-GCaMP6f, and stimulated with ionomycin following a previously published approach [32]. We stimulated wild-type and STIM1-KO cells, cultured in fetal bovine serum (FBS)-containing medium, with ionomycin and measured the extent of the increase in the fluorescence intensity relative to baseline (F/F0) of the GCaMP6-210, which then returned back to basal levels (Figure 4). The dynamics of the kinetics of F/F0 was similar in both wild-type and STIM1-KO cells (Figure 4b,c), and the maximal F/F0 values did not show any alteration in the absence of STIM1 (Figure 4d). Because the speed of mitochondrial Ca^2+^ import was not affected by the absence of STIM1 (Figure 4e), these data suggest that the mitochondrial maximal efficiency to import Ca^2+^ was not significantly altered by the lack of STIM1. In addition, no significant differences were observed in total levels of the mitochondrial Ca^2+^ uniporter (MCU) between wild-type and STIM1-KO cells, nor in the expression of the Grp75 (75 kDa glucose-regulated protein) and VDAC1 (voltage-dependent anion channel 1) (Figure 4f), two proteins that can associate with ITPR3 to facilitate the Ca^2+^ transfer between ER and mitochondria.

However, the steady-state [Ca^2+^]_mit_ was markedly decreased in STIM1-KO (Figure 4g), also reported previously by our group [24]. Notably, the wild-type phenotype was rescued by the overexpression of ITPR3 tagged with mCherry in STIM1-KO cells (Figure 4g). The level of ITPR3-mCherry compared to the endogenous ITPR3 is shown in Figure A4. As a control of this experiment, we assessed the [Ca^2+^]_mit_ in cells expressing the mCherry tag only (Figure 4h), an experiment that proved that the expression of this tag did not modify the rescue of the [Ca^2+^]_mit_ by the overexpression of ITPR3, as shown in Figure 4g.

We were able to recapitulate this phenotype in STIM1-KO cell lines stably overexpressing ITPR3-Myc (Figure 4i,j), where we assessed the [Ca^2+^]_mit_ of two positive clones (labeled as clones #29 and #57). Our data confirmed that stable overexpression of ITPR3 rescued the defects in [Ca^2+^]_mit_ triggered by the deletion of STIM1, which strongly suggests that ITPR3 deficiency was the cause of [Ca^2+^]_mit_ dysregulation in STIM1-KO cells.

### 2.4. Overexpression of ITPR3 Normalizes Mitochondrial Function

To understand whether the expression of ITPR3 could also rescue mitochondrial fitness in STIM1-KO cells, we measured the oxygen consumption rate (OCR) sensitive to oligomycin, carbonyl cyanide 4-(trifluoromethoxy)phenylhydrazone (FCCP), and rotenone + antimycin A, as a readout of the mitochondrial functionality and metabolic status (Figure 5). The metabolic assay indicated that basal respiration was lower in STIM1-KO cells, and that most of the O_2_ consumption was non-mitochondrial (Figure 5a). Indeed, the coupling efficiency of the electron transport and ATP synthesis was significantly lower in STIM1-deficient cells, as well as the overall ATP production.

Remarkably, the stable overexpression of ITPR3, which partially normalized [Ca^2+^]_mit_ in STIM1-KO cells, normalized basal oxygen consumption, coupling efficiency, and ATP production (Figure 5a), further confirming that ITPR3 is an essential regulator of mitochondrial function and suggesting that variations in ITPR3 levels constitute a major hallmark for STIM1-deficient cells. This latter conclusion was strengthened by the fact that the basal cell death observed in STIM1-KO cell cultures was prevented by the stable overexpression of ITPR3 in these cells (Figure 5b).

## 3. Discussion

As a major Ca^2+^ signaling regulator, STIM1 is involved in a number of pathologies, including neurodegenerative diseases, severe immunodeficiency, and cancer [33,34,35,36]. In primary cancer samples, as well as in cancer cell lines, the total amount of STIM1 correlates with augmented migration and proliferation rates, and consequently with poor prognosis (reviewed in Reference [37]). Conversely, the decline in total STIM1 levels in brain tissue of Alzheimer’s disease patients correlates with the Braak stage [24], again suggesting a role for STIM1 in cell survival.

In order to understand the molecular mechanism by which STIM1 regulates cell viability in excitable cells, we have examined here STIM1-KO SH-SY5Y cells as an in vitro cellular model. In this system, the deficiency in STIM1 does not affect differentiation to a neuronal-like cell type, but it does trigger a significant loss of cell viability [24]. Using this system, we searched for alterations in the expression of Ca^2+^ regulators and transporters, finding a strong decrease in *ITPR3* transcripts and ITPR3 protein expression in STIM1-deficient cells, as is reported here. ITPRs are most abundantly expressed in the ER, but they are also found in the nuclear envelope, Golgi apparatus, secretory vesicles, and plasma membrane (reviewed in References [38,39]). The significant decrease in ITPR3 is particularly important since an enrichment of ITPR3 in mitochondrial-associated membranes from SH-SY5Y cells is reported in this work. Similar distribution of ITPR3 has been reported for other cells lines [27,29,40], and it is assumed that this localization in MAMs promotes the shuttling of Ca^2+^ between ER and mitochondria [41,42].

ER-mitochondria Ca^2+^ transfer is essential for mitochondrial function because Ca^2+^ positively regulates three enzymatic reactions of the Krebs cycle (isocitrate dehydrogenase, α-ketoglutarate dehydrogenase, and pyruvate dehydrogenase) [43], stimulating the supply of NADH, and therefore, normalizing the mitochondrial electron transport chain. In this regard, it has been reported that the deficiency of Ca^2+^ transfer from the ER to the mitochondria triggers autophagy as a result of the failure in the energy production [44]. On the contrary, mitochondrial Ca^2+^ overload has been related to the opening of the mitochondrial permeability transition pore, which ultimately leads to other types of cell death [45,46].

The need for a constitutive Ca^2+^ transfer between ER and mitochondria has been reported in several human tumorigenic cell lines [47], and the relationship between ITPR3 expression or activity and cell survival is particularly critical in the case of cancer cells. For instance, in colon cancer, greater ITPR3 expression is associated with cancer cell proliferation and lower 5-year survival of patients. On the other hand, *ITPR3* knock-down in Caco-2 colon cancer cells enhances apoptosis, while over-expression enhances cell survival [48]. Similarly, cholangiocarcinoma cells were shown to be particularly sensitive to the knock-out of *ITPR3*, a genetic manipulation that led to the reduction of cell migration, as well as to the reduction of mitochondrial oxygen consumption and proliferation [49]. The migration and invasion potential have been shown to be severely influenced by ITPR3 levels in other cancer cells. Whereas high levels of ITPR3 were observed in the highly migrating and invasive MDA-MB-231 and MDA-MB-435S breast cancer cell lines, the low-migrating MCF-7 showed low levels of ITPR3, but the stable overexpression of ITPR3 increased the migration capacity of this cell line [50]. In colorectal cancer DLD-1 cells, the silencing of ITPR3 expression led to a reduction of tumor volumes after injection of these cells in nude mice, increasing the apoptosis of these cells in hypoxic conditions and demonstrating the proliferative and anti-apoptotic role of ITPR3 [51]. Therefore, the direct correlation between STIM1 protein expression level and ITPR3 expression found here may explain the suggested correlation between STIM1 protein levels and proliferation and malignancy in cancer cells [37,52,53].

In the specific case of neuronal cells, some mutations in PSEN1/2 linked to Alzheimer’s disease leads to the partial inhibition of presenilins/γ-secretase activity, a result that end up in upregulated MAM function and ER-mitochondria communication [54]. The functional consequence for this upregulation is the altered Ca^2+^ homeostasis and altered cholesterol and phospholipid metabolism. However, in the absence of these mutations, like in the case reported here, the mitochondrial Ca^2+^ homeostasis and bioenergetics were significantly altered by the absence of STIM1 and the subsequent reduction of ITPR3. It must be emphasized also that the ITPRs are positive regulators of MAMs and that the expression of any of these isoforms on triple KO cells (knock-out for the 3 ITPR variants) increased the number of ER-mitochondria contacts and that ITPR2 and ITPR3 were the two most effective isoforms engaging ER-mitochondria Ca^2+^ transfer [55].

On the other hand, STIM2, a STIM1 paralogue, performs similar but not identical functions. For example, the dissociation constant of STIM2 for Ca^2+^ is close to 0.5 mM [56], which is significantly higher than that of STIM1 which is around 0.25 mM [6]. Therefore, the function of STIM2 is closely related to the maintenance of a basal intraluminal Ca^2+^ concentration, while STIM1 acts in response to more intense variations of this concentration. The transmembrane domain of both proteins is very similar, so it is possible that STIM2 is also a substrate of gamma-secretase activity, as suggested elsewhere [20]. In fact, it has been described that STIM2 abundance is reduced in cells expressing some FAD-associated variants of PSEN1 [23,57], and that STIM2 reduction destabilizes mature dendritic spines in mice bearing FAD-associated PSEN1 variants [57]. Therefore, it would be necessary to address the study of the role of STIM2 in this neuronal model (SH-SY5Y) by means of total gene suppression, as we have carried out here with CRISPR/Cas9-mediated gene editing for STIM1.

The present results in SH-SY5Y demonstrate that the reduced ITPR3 levels observed in STIM1-deficient cells are responsible for the reduced Ca^2+^ release from the ER in response to the activation of the phosphoinositide pathway. Because the Ca^2+^ import capability of mitochondria in STIM1-KO is not significantly deregulated, our results suggest that the decrease of ITPR3 is also responsible for the reduction in [Ca^2+^]_mit_. This conclusion is supported by the normalization of [Ca^2+^]_mit_, as well as the recovery of the basal OCR, ATP production, and cell viability, observed in STIM1-KO cells overexpressing ectopic ITPR3. For this reason, our results indicate that STIM1 has a key role in the normalization of [Ca^2+^]_mit_ by controlling ITPR3 protein levels. Interestingly, postmortem cerebella from patients with the PSEN1-E280A mutation, typical of FAD, showed much lower levels of ITPR3 compared to control patients, and the expression of PSEN1-E280A in a neuronal cell line altered ER-mitochondria tethering compared with that in cells expressing wild-type PSEN1 [58]. Because we also have reported a significant decrease of STIM1 in brain samples from SAD patients [24], the axis STIM1-ITPR3-[Ca^2+^]_mit_ must be considered to explain the modulation of neurodegeneration and survival.

## 4. Materials and Methods

### 4.1. Chemicals

SH-SY5Y cells were purchased from ECACC (European Collection of Authenticated Cell Cultures) and distributed by Sigma-Aldrich (St. Louis, MO, USA); *all-trans*-retinoic acid (RA), collagen type I solution (Ref. #C3867), Dulbecco’s modified Eagle’s medium (DMEM), DMEM:F-12 Ham’s medium, and carbachol (carbamoylcholine chloride) were purchased from Sigma-Aldrich. Fura-2-acetoxymethyl ester (fura-2-AM) was from Merck Millipore (Darmstadt, Germany); Clarity Max™ Western ECL substrate was from Bio-Rad (Hercules, CA, USA).

### 4.2. Antibodies

The rabbit polyclonal anti-STIM1 antibody (#4119) was from ProSci Inc. (Poway, CA, USA); the mouse monoclonal anti-red fluorescent protein (RFP, clone 6G6) was from Chromotek (Planegg-Martinsried, Germany); the mouse anti-beta tubulin antibody (clone TUB2.1) was from Sigma-Aldrich; the mouse anti-ITPR3 antibody (#610312), was from BD Biosciences (Franklin Lakes, NJ, USA); the mouse anti-ITPR2 antibody (sc-398434, clone A-5), the mouse anti-ITPR1/2/3 antibody (sc-377518, clone B-2), the mouse anti-ASCL4 (sc-365230, clone F-4), mouse anti-VDAC1 (sc-390996, clone B-6), and the mouse monoclonal anti-GAPDH antibodies were from Santa Cruz Biotechnology (Heidelberg, Germany); the rabbit anti-ITPR1 antibody (RBT-03) was a kind gift from Dr. Jan B. Parys (KU Leuven) who generated and characterized the antibody elsewhere [59]. The anti-calreticulin (ab16144) was from AbCam (Cambridge, UK), and the rabbit monoclonal anti-MCU (#14997) and the rabbit monoclonal anti-Grp75 (#3593) were from Cell Signaling Technology (Danvers, MA, USA); Secondary horseradish peroxidase (HRP)-labeled antibodies were from Pierce (ThermoFisher Scientific, Waltham, MA, USA).

### 4.3. DNA Constructs

The construct for the expression of ITPR3-mCherry was made by inserting the human *ITPR3* cDNA (NM_002224.4) into the pmCherry-N1 vector using the cDNA from the Harvard plasmid clone HsCD00399229 as a template. The following primers were used to insert *Sma*I flanking sites (underlined): ITPR3-fwd: 5′-TCCCCCGGG-ATGAGTGAAATGTCCAGCTTTC, and ITPR3-rev: 5′-TCCCCCGGG-GGCGGCTAATGCAGTTCTGGAC. The construct for the expression of mouse Itpr3-Myc was purchased from OriGene (Rockville, MD, USA) (clone #MR225699).

Transfection of cells with DNA constructs was performed with 1–2 μg plasmid DNA per 10-cm dish and polyethylenimine (Polysciences Inc., Eppelheim, Germany) in serum-containing medium.

### 4.4. Culture, Differentiation and Generation of Stable Cell Lines

Cells were cultured in Dulbecco’s modified Eagle’s medium (DMEM) with 10% (*v/v*) heat-inactivated fetal bovine serum (FBS) (Ref. 10500-064 from ThermoFisher Scientific), 2 mM L-glutamine, 100 U/mL penicillin, and 0.1 mg/mL streptomycin in a humidified atmosphere of 95% air/5% CO_2_ at 37 °C, as described elsewhere [60]. Cell culture dishes and glass coverslips were treated with 1.5 g/mL collagen type I solution (freshly diluted in Milli-Q water from the stock solution purchased from Sigma). Collagen treatment was extended for a minimum of 30 min at 37 °C in a humidified atmosphere to avoid drying, and then it was washed 3 times with PBS before cell plating. The differentiation of cells was performed as described previously [24]. Stable cell lines overexpressing ITPR3-Myc were generated by selection with G418 (0.5 mg/mL) for 7 days in culture, and subsequent analysis of individual clones.

The generation of the SH-SY5Y STIM1-KO cell line by genome editing with CRISPR/Cas9 is described elsewhere [24].

Mycoplasma contamination was monitored with the Mycoplasma Gel Detection Kit (Ref. #4542) from BioTools (Madrid, Spain).

### 4.5. Quantitative Real-Time PCR

RNA was isolated using the Quick-RNA MiniPrep Plus kit (Zymo Research, Irvive, CA, USA), and the cDNA was generated from 1 μg of total RNA using High Capacity cDNA Reverse Transcription kit (Applied Biosystems, Foster City, CA, USA). To evaluate levels of *ITPR* transcripts we used SYBR Green-based quantitative PCR (qPCR). The sequences of the primers were as follows: *ITPR1*-fwd: 5’-CTGCCACCAGTTCAAAAGCC and *ITPR1*-rev: 5’-CCACCTCTGCTGCCAAGTAA to detect *ITPR1* transcript variants 1, 2, and 3 (NM_001099952.2, NM_002222.5, NM_001168272.1); *ITPR2*-fwd: 5’-GCAGGGAAGAAGAGGGACG and *ITPR2*-rev: 5’-ACCCCAAGGTGCTGATGAAG to detect *ITPR2* transcripts (NM_002223.4); *ITPR3*-fwd: 5’-TATGCAGTTTCGGGACCACC and *ITPR3*-rev: 5’-TGCCCTTGTACTCGTCACAC to detect *ITPR3* transcripts (NM_002224.4). Expression was normalized to ribosomal protein L32 transcript variant 1, *RPL32* (NM_000994.3), using the following primers: *RPL32*-fwd: 5’-CATCTCCTTCTCGGCATCA and *RPL32*-rev: 5′-CTGGGTTTCCGCCAGTTAC (amplicon size 155 bp). Technical triplicates were performed from 2 biological replicates (n = 6). Real-time PCR was performed using a QuantStudio 6-Flex (Applied Biosystems). Data were analyzed using the QuantStudio Real-Time PCR software (ThermoFisher Scientific). Other transcripts were evaluated with the Human Intracellular Calcium Signaling TaqMan Array 96-well FastPlate (Cat. #4418932, Applied Biosystems), using *GAPDH*, *HPRT1*, and *GUSB* as housekeeping transcripts for the normalization.

### 4.6. Lysis of Cells and Immunoblot

Cells were lysed in the following buffer: 50 mM Tris-HCl (pH 7.5), 1 mM EGTA, 1 mM EDTA, 1% (*w/v*) Nonidet P40, 1 mM sodium orthovanadate, 50 mM NaF, 5 mM sodium pyrophosphate, 0.27 M sucrose, 1 mM DTT, 1 mM benzamidine, and 0.1 mM phenylmethylsulfonyl fluoride. The last three reagents of this list were added from a stock solution just before the use of the lysis buffer. Clarification of samples was performed after lysis with 0.75–1 mL of ice-cold lysis buffer/dish and centrifugation at 4 °C for 15 min at 20,000× *g*. Samples were sonicated with 5 × 10 s pulses with a setting of 45% amplitude using a Branson Digital Sonifier (ThermoFisher Scientific). Protein concentration was determined using the Coomassie Protein Assay Reagent (ThermoFisher Scientific).

Lysates (10–40 μg) were subjected to electrophoresis on polyacrylamide gels and subsequent electroblotting to nitrocellulose membranes. Membranes were blocked for 1 h at room temperature (RT) in blocking buffer: TBS-T (Tris-buffered saline buffer, pH 7.5, with 0.2% Tween-20) containing 10% (*w/v*) non-fat milk. Then the membranes were incubated overnight with the specific antibody diluted in blocking solution at 4 °C, washed with TBS-T, and then incubated with anti-IgG horseradish peroxidase (HRP)-conjugated secondary antibodies (1:10 000 dilution in all cases) for 1 h at RT. Dilution of antibodies were as follows: anti-STIM1 antibody (1 μg/mL), anti-ITPR1 (1/1000 dilution), anti-ITPR2 (0.4 μg/mL), anti-ITPR3 (0.25 μg/mL), anti-ITPR1/2/3 (0.4 μg/mL), anti-RFP (clone 6G6, 1/1000 dilution), anti-beta tubulin (clone TUB2.1, 1/3000 dilution), and anti-GAPDH (1/3000 dilution). Clarity Max™ Western ECL substrate was added to the membranes and the signal recorded with ChemiDoc XRS+ system (BioRad, Hercules, CA, USA). The recorded signal was quantified by volumetric integration using the Image Lab software (BioRad).

### 4.7. XF Cell Mito Stress Assay

Cell bioenergetics was evaluated in a Seahorse XFp analyzer (Agilent Technologies, Santa Clara, CA, USA). SH-SY5Y cells were plated on Seahorse XFp plates (55,000 cells/per well). After 24 h, cells were washed with XF DMEM medium followed by a 1 h incubation at 37 °C in a CO_2_-free incubator. Each plate contained two wells without cells to serve as blank controls. After monitoring basal respiration, cells were sequentially treated with 1.5 μM oligomycin, 1 μM FCCP, and 0.5 μM antimycin A + 0.5 μM rotenone (Seahorse XFp Cell Mito Stress kit, from Agilent Technologies). Oxygen consumption rate (OCR) data were normalized to the total cell amount per well estimated by Janus Green staining [61]. Briefly, after bioenergetics analysis, cells were fixed in 4% paraformaldehyde (PFA) and incubated with 0.2% Janus Green B in PBS for 3 min at room temperature. The excess of dye was removed by dipping into cold water and gentle shaking, and the bound dye was dissolved in 0.5 N HCl (0.1 mL/well) and the optical density at 595 nm was evaluated. The metabolic parameters of the assay were calculated with the Seahorse Wave software. In all cases, 3 technical replicates were recorded in every experiment, and the experiments were carried out with 3 biological replicates.

### 4.8. Isolation of Mitochondria-Associated ER Membranes

The isolation of MAMs was performed as described previously [62]. Briefly, cells from 50–60 10-cm dishes were scrapped in cold PBS and concentrated by centrifugation at 3000× *g* for 15 min. Cells were then washed in the following homogenization buffer (HB): 225 mM mannitol, 25 mM HEPES-KOH (pH = 7.4), 1 mM EGTA, and protease inhibitor cocktail. Cells were homogenized with 8 strokes in a Teflon-pestle grinder. The homogenate was clarified with 2 sequential centrifugations at 600× *g* for 10 min. The resulting supernatant was then centrifuged at 10,000× *g* to pellet the mitochondrial fraction. This pellet was resuspended in 1 mL of HB and loaded onto a freshly prepared 30% Percoll solution in HB and then centrifuged at 95,000× *g* for 30 min. The lower-density upper band was collected with the tip of a Pasteur pipette, diluted in 5 volumes of HB, vortexed for 15 s, and centrifuged twice at 8000× *g* for 10 min to remove the contaminant Percoll. The supernatant was centrifuged for 1 h at 100,000× *g* and the floating clear membrane pellet was collected as the pure MAMs fraction. All centrifugations were carried out at 4 °C. Other details are described in Reference [62].

### 4.9. Cytosolic, Mitochondrial, and ER Free Calcium Concentration

Cytosolic free calcium concentration ([Ca^2+^]_i_) was measured in fura-2-AM-loaded cells as described elsewhere [24,63,64,65,66]. Excitation fluorescence wavelengths were selected with 340/26 and 387/11 nm filters (Semrock, Rochester, NY, USA), and emission fluorescence with a 510/10 nm filter. Calibration of the fura-2 ratio signal was performed as indicated in Reference [24]. Cells were stimulated with 100 μM carbachol in Ca^2+^-free HBSS with the following composition: 138 mM NaCl; 5.3 mM KCl; 0.34 mM Na_2_HPO_4_; 0.44 mM KH_2_PO_4_; 4.17 mM NaHCO_3_; 4 mM Mg^2+^; EGTA 0.1 mM (pH = 7.4). All experiments were carried out at 36 °C.

Mitochondrial free calcium concentration ([Ca^2+^]_mit_) was measured as described elsewhere [24,67]. Basically, [Ca^2+^]_mit_ was assessed in cells transfected with the genetically-encoded Ca^2+^ sensor 4mtD3cpv. CFP, YFP, and FRET efficiency between the two channels was measured with the dual CFP/YFP-2×2M-B filter set (Semrock). All measurements were performed in Ca^2+^-containing HBSS for 4–5 min. Spectral unmixing (i.e., subtracting the bleed-through from one channel into another) was performed by determining the bleed-through coefficients as described in References [24,67]. The background corrected ratio, i.e., Ratio = (FRET_ROI_ − FRET_background_)/(CFP_ROI_ − CFP_background_) was converted to [Ca^2+^]_mit_ as described in Reference [67], using a dissociation constant for Ca^2+^ = 0.76 μM. The FRET/CFP ratio (R) was evaluated after calibrating the signal with the subsequent addition of 5 μM ionomycin + 5 mM EGTA (Rmin), followed by the addition of 5 μM ionomycin + 10 mM Ca^2+^ (Rmax), as in Reference [24].

To study maximal Ca^2+^ uptake by mitochondria, [Ca^2+^]_mit_ dynamics were assessed in cells transfected with the genetically-encoded Ca^2+^ sensor mito^4×^-GCaMP6f, due to its improved dynamic range, which results in a much better signal-to-noise ratio when detecting peak responses, as described in Reference [68]. This probe was designed to express 4 consecutive copies of the signal peptide of COX8 (MSVLTPLLLRGLTGSARRLPVPRAKIHSLGDP) and a short linker (RSGSAKDPT) before the sequence of GCaMP6f [68]. The main advantage of this probe is that it shows increased response rate compared to 4mtD3cpv, allowing the fast monitoring of the Ca^2+^ uptake by mitochondria. All measurements were performed in L15 medium + 10% FBS and temperature was controlled and set to 36 °C. Excitation fluorescence wavelengths were selected with a 485/10 filter (Semrock), and emission fluorescence with a 535/20 nm filter.

Endoplasmic reticulum free calcium concentration ([Ca^2+^]_ER_) was measured using the genetically-encoded Ca^2+^ sensor ER-GCaMP6-210 [31]. Transfected cells were monitored using the following settings: excitation fluorescence wavelengths were selected with 480/30 nm filters (Semrock), and emission fluorescence with a 535/40 nm filter. The emission of fluorescence was calibrated with the subsequent addition of 5 μM ionomycin + 5 mM EGTA followed by the addition of 5 μM ionomycin + 10 mM Ca^2+^.

All measurements were performed using an EM-CCD C9100 digital camera (Hamamatsu Photonics, Hamamatsu City, Japan) attached to a Nikon Eclipse Ti-E inverted microscope (Nikon Instruments Europe B.V., Amsterdam, The Netherlands). Illumination was performed with a xenon arc lamp. All measurements were performed in a DH-35iL culture dish incubator, and the temperature was set at 36 °C (Warner Instruments, Holliston, MA, USA). In all cases, excitation and emission conditions were controlled by the NIS-Elements AR software.

### 4.10. Cell Death Analysis

Cells were plated on 8-well μslide plates from Ibidi (Gräfelfing, Germany) at 40,000 cells/well density. Forty-eight hours after plating, cells were stained with the Live/Dead Cell Imaging kit (ThermoFisher Scientific), according to the manufacturer protocol. Ten images from randomly chosen fields were recorded for every experimental condition. Imaging was performed with a 20× objective on a Nikon Eclipse Ti-E inverted microscope.

### 4.11. Statistical Analysis of Data

Statistical analyses between pairs of data groups were done using the Mann–Whitney test of data (non-parametric unpaired *t*-test). Analyses were performed with the GraphPad software. Differences between groups of data were taken statistically significant for *p* < 0.05. The *p*-values are represented as follows: (*) *p* < 0.05, (**) *p* < 0.01, and (***) *p* < 0.001.

## Figures and Tables

**Figure 1 ijms-21-06598-f001:**
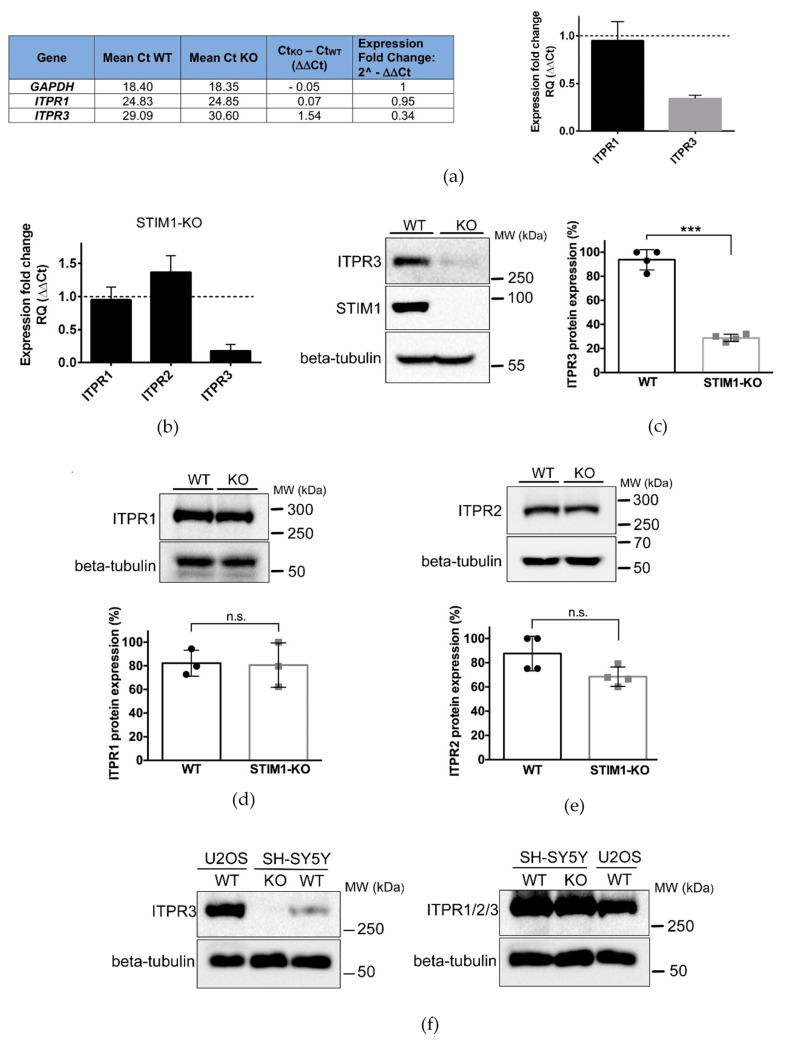
Downregulation of ITPR3 in STIM1-deficient cells. (**a**) Total RNA was purified from undifferentiated wild-type (WT) and STIM1-KO cells, and the quantification of transcripts was evaluated from a Taqman Gene Expression Array (Ref. #4418932). Data from 3 different experiments are given in the Appendix A (n = 3 for WT; n = 3 for KO). From these data, the threshold cycle (Ct) was calculated to evaluate the expression fold change as 2^(−ΔΔC*t*) which is also plotted in the bar chart, as the fold-change of *ITPR1/3* expression in STIM1-KO cells compared with wild-type cells. Expression of GAPDH was used as a housekeeping gene. (**b**) Quantification of *ITPR1/2/3* transcripts was performed individually with specific primers (see Methods, Section 4.5). The bar chart depicts the expression fold change of transcripts in STIM1-KO cells compared with wild-type cells. Data are mean ± S.D. from 2 different biological replicates with technical triplicates (n = 6). (**c**–**e**) ITPR3, ITPR1, and ITPR2 protein expression in whole cell lysates was quantified in wild-type and STIM1-KO cells. Beta-tubulin was used as a loading control. Data are mean ± S.D. from 3 independent experiments (for ITPR1) or 4 independent experiments (for ITPR2 and ITPR3). (**f**) ITPR3 protein expression (left panel) or total levels of ITPRs (right panel) were evaluated in osteosarcoma U2OS cells, as well as in wild-type and STIM1-KO SH-SY5Y cells by immunoblot. Beta-tubulin was used as a loading control. (***) Statistical significance *p* < 0.001.

**Figure 2 ijms-21-06598-f002:**
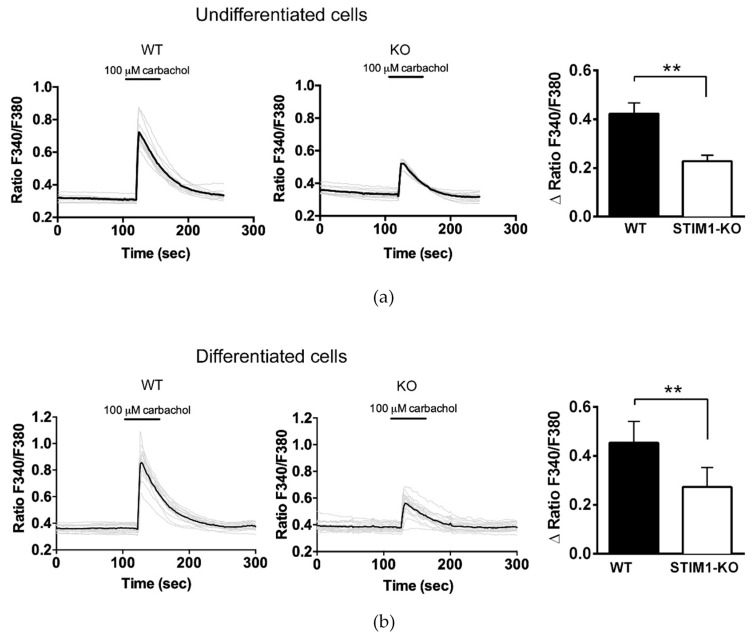
Decrease of carbachol-evoked Ca^2+^ release in STIM1-deficient cells. Undifferentiated cells (**a**) or differentiated cells (**b**) were loaded with fura-2 to measure Ca^2+^ kinetics in response to a 30 s pulse of 100 μM carbachol (CCh) in Ca^2+^-free medium. After CCh addition, cells were washed out 3 times with Ca^2+^-free Hank’s balanced salt solution (HBSS). The increase of the ratio F340/F380 induced by CCh was quantified from a minimum of 40 cells and 4 independent experiments (n = 68 undifferentiated WT cells, n = 58 undifferentiated KO cells, n = 53 differentiated WT cells, n = 40 differentiated KO cells). The results are shown in the bar chart as mean ± S.D. (**) Statistical significance *p* < 0.01.

**Figure 3 ijms-21-06598-f003:**
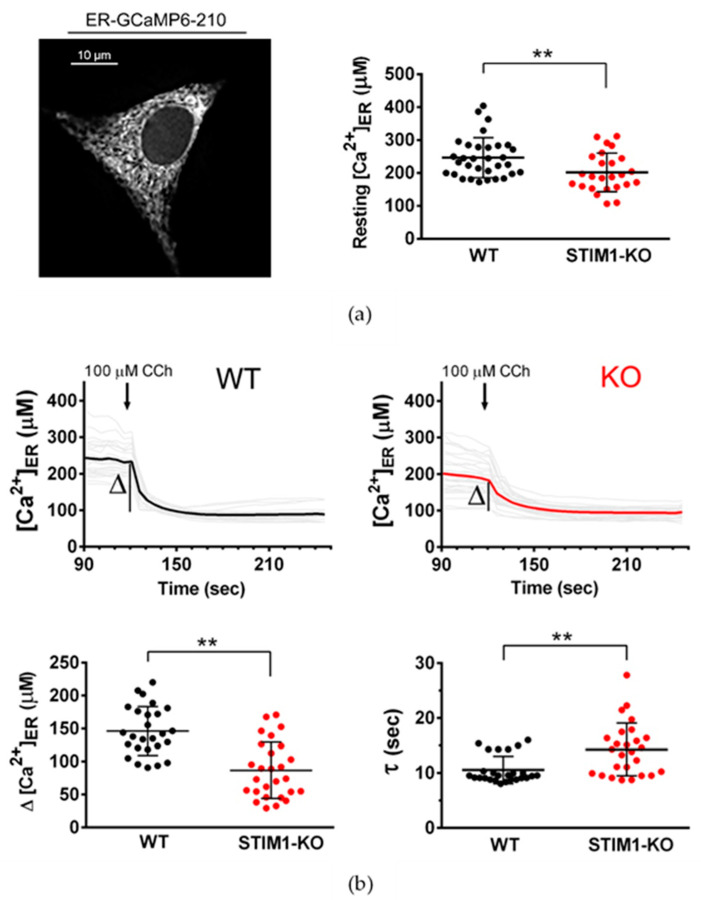
Free Ca^2+^ concentration within the endoplasmic reticulum (ER) and Ca^2+^ release triggered by CCh. (**a**) The ER-specific Ca^2+^ sensor ER-GCaMP6-210 was used to measure steady-state levels of [Ca^2+^]_ER_ in undifferentiated SH-SY5Y cells. The figure shows the distribution of the sensor under excitation wavelength with 480 nm (Emission = 510 nm). Forty-eight hours after transfection, the [Ca^2+^]_ER_ was evaluated for cells incubated in Ca^2+^-containing HBSS. Data are mean ± S.D. from 2 independent experiments (n= 32 ROIs from 13 WT cells, and n= 25 ROIs from 13 KO cells). (**b**) Transfected cells with ER-GCaMP6-210 were stimulated with 100 μM CCh and the amplitude of the decrease in [Ca^2+^]_ER_ and time constant tau (τ) of the kinetics of [Ca^2+^]_ER_ in response to CCh were calculated. Data are mean ± S.D. from 13 cells and 26 ROIs per experimental condition. (**) Statistical significance *p* < 0.01.

**Figure 4 ijms-21-06598-f004:**
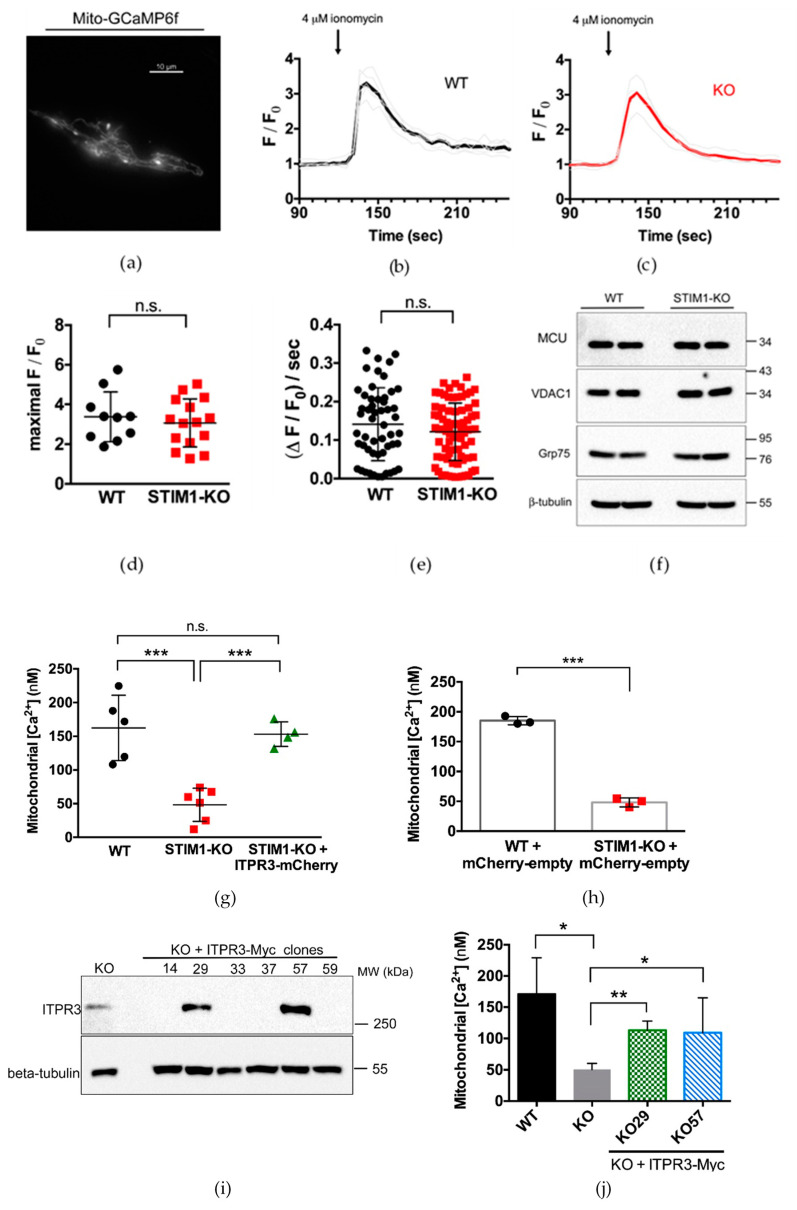
ITPR3 overexpression restores [Ca^2+^]_mit_. (**a**) The mitochondria-specific Ca^2+^ sensor mito^4×^-GCaMP6f was used to measure mitochondrial [Ca^2+^] dynamics in undifferentiated cells in response to a stimulation with ionomycin. The figure shows the distribution of the sensor under 480 nm excitation wavelength (Emission = 510 nm). (**b**,**c**) Forty-eight hours after transfection, cells were transferred to a bicarbonate-free L15 medium + 10% fetal bovine serum (FBS) and stimulated with a single pulse of 4 μM ionomycin. Time course of emission of fluorescence (F) was recorded for 5 min. Single traces (grey lines) are different ROIs from the same experiment. Colored lines (black for WT and red for KO cells) depict the average F/F_0_ signal over the time. Plots are representative of >10 independent experiments (n = 51 ROIs for WT, n = 78 ROIs for KO). (**d**) The maximal F/F_0_ ratio attained after ionomycin addition was calculated and the results are shown as a scatter plot. (**e**) Slope of the increase of F/F_0_ ratio during the first 20 s after the addition of ionomycin. Data are from traces in panels (**b**,**c**). (**f**) Wild-type and STIM1-KO SH-SY5Y cells were assessed for the expression of the mitochondrial Ca^2+^ uniporter (MCU), VDAC1, and Grp75 by immunoblot. Beta-tubulin was used as a loading control. (**g**) [Ca^2+^]_mit_ was evaluated in cells transfected for the transient expression of 4mtD3cpv. In all cases, the ratio of the fluorescence resonance energy transfer (FRET) over the cyan fluorescent protein (CFP) signal, (FRET/CFP ratio) was recorded from cells in Ca^2+^-containing HBSS. The ratio signal was calibrated, and the data shown are mean ± S.D. from 3 independent experiments. (**h**) WT and STIM1-KO cells were transfected with the empty mCherry vector that served as a backbone for the ITPR3-mCherry constructs expressed in panel G, and [Ca^2+^]_mit_ was evaluated with the 4mtD3cpv sensor. (**i**) STIM1-KO cells were transfected for the stable expression of ITPR3-Myc. G418-resistant clones were assessed for ITPR3 overexpression compared with untransfected STIM1-KO cells (KO). Although G418 selection yielded several clones, only two of them (#29 and #57) were positive for ITPR3-Myc expression. (**j**) Positive clones stably overexpressing ITPR3-Myc (clones #29 and #57) were transfected with 4mtD3cpv to evaluate [Ca^2+^]_mit_ from cells in Ca^2+^-containing HBSS. Data are mean ± S.D. from 3 independent experiments. Statistical significance: (*) *p* < 0.05, (**) *p* < 0.01, and (***) *p* < 0.001.

**Figure 5 ijms-21-06598-f005:**
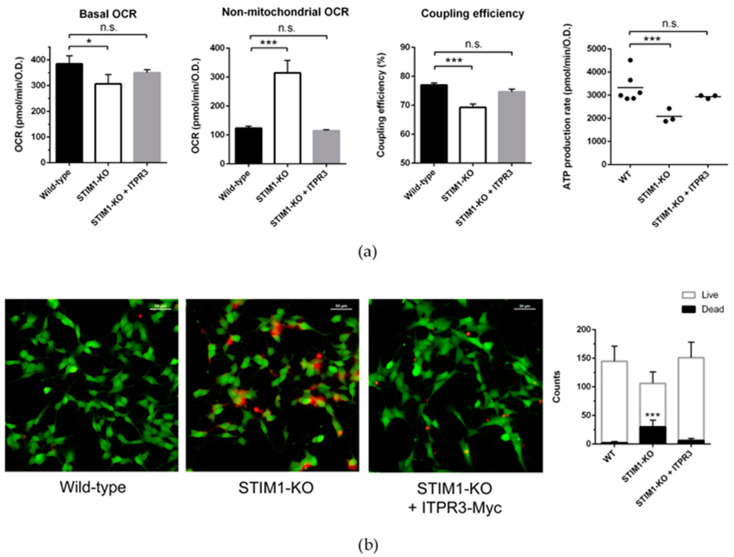
Overexpression of ITPR3 normalizes mitochondrial bioenergetics and cell survival in undifferentiated STIM1-KO cells. (**a**) Bioenergetics analysis was performed with a Seahorse XFp Analyzer (Agilent Technologies, Santa Clara, CA, USA), as indicated in the Methods section. (**b**) Cells were plated on μslide plates (Ibidi) and stained with Live/Dead Cell Imaging Kit (ThermoFisher Scientific, Waltham, MA, USA). Representative images of the experimental conditions under study are shown. Quantification of live/dead cells was performed from 10 randomly chosen fields, and the average number of cells per frame is plotted as mean ± S.D. Non-parametric unpaired *t*-test result for dead cells count is indicated: (*) *p* < 0.05 and (***) *p* < 0.001.

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
