# Peer review of "STIM1 Deficiency Leads to Specific Down-Regulation of ITPR3 in SH-SY5Y Cells"

_ijms, 2020, doi:10.3390/ijms21186598_

Round 1
Reviewer 1 Report
This manuscript by Pascual-Caro et al addresses the role of STIM1 in regulation of IP3 receptor-3 expression and the Ca2+ pathways downstream of IP3R3: the mitochondrial free Ca2+ concentration ([Ca2+]mit) and ATP synthesis. Using neuroblastoma SH-SY5Y cell line as a model, the authors showed that knocking-out of STIM1 gene by genome editing with CRISPR/Cas9 resulted in a significant reduction of IP3R3 expression, which was associated with lower steady state [Ca2+]mit, lower oxygen consumption, lower rate of ATP production and reduced cell viability. The ectopic expression of IP3R3 in STIM1-KO cells seemed to have restored basal levels of [Ca2+]mit and normalised mitochondrial function. The results look convincing, however, there is a question about the calibration of the data presented in Fig 4 g, h.
Specifically:
It is not clear, how basal level of [Ca2+]mit in STIM1-KO cells can drop as low as ~50 nM, which is well below the expected level of free Ca2+ concentration in the cytosol. Data presented in Fig 2 suggests that STIM1-KO does not affect steady state [Ca2+]cyt, and data in Fig 4f indicate that STIM1-KO does not affect Ca2+ transport mechanism into mitochondria. This implies that steady state [Ca2+]mit cannot go below [Ca2+]cyt (normally between 100 and 200 nM). The authors need to determine the resting level of [Ca2+]cyt in SH-SY5Y cells and make sure that calibration of the Ca2+ signals in mitochondria is consistent with the Ca2+ levels in the cytosol. Alternatively, they need to explain how reduction in IP3R3 expression can drive [Ca2+]mit below the resting levels of [Ca2+]cyt.
Reviewer 2 Report
Reviewers’ comments to : “STIM1 Deficiency Leads to Specific Down-Regulation of ITPR3 in SH-SY5Y Cells” by Pascual-Caro et al. (ijms-920897)
In this work Authors investigated the effect of the artificial modulation of the expression of the STIM1 protein, a key component of the capacitative calcium influx machinery on the expression of the type 3 inositol trisphosphate receptor calcium channel (IP3R3, coded by the ITPR3 gene), and its consequences on mitochondrial function, in the SH-SY5Y human neuroblastoma cell line, a widely accepted model of in vitro neural differentiation. Using STIM1 knock-out in SH-SY5Y cells, Authors show convincingly that STIM1 deficiency leads to significantly decreased IP3R3 expression, whereas the expression of the other IP3R isoforms (IP3R1 and IP3R2) is not modified significantly. It is also shown that IP3R3 is enriched at endoplasmic reticulum-mitochondrial contact sites. This is followed by functional studies in which Authors show that IP3R3 downregulation in STIM1-deficient cells leads to decreased IP3-dependent calcium mobilization from the endoplasmic reticulum in undifferentiated, as well as all-trans-retinoic acid-differentiated SH-SY5Y cells, and decreased resting cytosolic and intra-ER luminal calcium levels and slower calcium mobilization are reported as well. Investigation of mitochondrial calcium homeostasis disclosed lower mitochondrial resting calcium levels in STIM1 knock-out cells, and decreased mitochondrial function, including oxygen consumption and ATP synthesis, as well as a generally decreased cellular fitness and enhanced apoptotic potential are also shown. Mitochondrial function and cell viability could be normalized by IP3R3 overexpression in STIM1-deficient cells.
Overall, Authors conclude that in SH-SY5Y cells STIM1 deficiency leads to decreased IP3R3 expression and slower IP3-induced calcium release, and consequently to impaired mitochondrial calcium homeostasis and function, leading to decreased cellular fitness and survival.
Decreased STIM1 expression has been found earlier in brain tissue in Alzheimer’s disease by the Authors. The main interest of the present work is the identification of the involvement of IP3R3 as a molecular actor downstream of STIM1 in this pathology. It is proposed that decreased IP3R3 expression, induced by deficient STIM1 expression, may lead to decreased mitochondrial calcium signaling and ATP production, and consequently, to decreased neuronal fitness in this disease. This constitutes an interesting new hypothesis that may contribute to the better understanding of the molecular pathogenesis of Alzheimer’s disease.
Comments:
- Authors state that IP3R3 is a minor IP3R isoform in SH-SY5Y cells (Fig. 1 and Lane 138). However, calcium fluorescence experiments indicate rather significant effects in STIM1 knock-out cells, in which IP3R3 protein levels are down-regulated (Fig. 2). How can this discrepancy be reconciled? Is there a preferential calcium mobilization at endoplasmic reticulum-mitochondrial contact sites despite the presence of the other IP3R isoforms in these cells?
- It is clearly shown in this work that STIM1 deficiency leads to IP3R3 down-regulation. However, no molecular mechanism for this phenomenon is proposed or investigated.
- It is shown in this work that STIM1 knock-out leads to decreased carbachol-induced calcium mobilization from the ER in proliferating, as well as all-trans-retinoic acid-differentiated SH-SY5Y cells (Fig. 2). Considering the direct involvement of STIM1 in the induction of capacitative calcium influx, it would be interesting to know also, what are the effects of STIM1 on this phenomenon. This could be easily measured in the experimental format presented in Fig. 2, by the re-addition of calcium into the medium, for example at the 250 sec. time points. If capacitative calcium influx is significantly impaired, this may help to explain the cytosolic, as well as endoplasmic reticulum and mitochondrial calcium depletion reported in this work. Is it possible that the various defects of cellular calcium homeostasis reported in this work are, at least in part, due to a partial calcium depletion of the cells, because a major calcium influx mechanism (capacitative calcium influx) has been incapacitated by STIM1 knock-out? Is IP3R3 down-regulation in the cells is caused by lower (cytosolic or endoplasmic reticulum luminal) calcium levels ? Could IP3R3 expression be rescued in STIM1 knock-out cells by calcium ionophore (ionomycin, A23187) or SERCA-inhibitor (thapsigargin, cyclopiazonic acid or 2,5-di-tert-butyl-1,4 benzo-hydroquinone) treatment ?
- Authors show that STIM1 expression is decreased in brains in Alzheimer’s disease, and show, in this work, that STIM1 deficiency leads to decreased IP3R3 levels. Are IP3R3 levels decreased in the brain in Alzheimer’s disease?
- Authors show in Fig. 4, Panel i various cell clones made from STIM1 knock-out SH-SY5Y cells, in which IP3R3 expression is reintroduced by transfection of an IP3R3-Myc fusion protein followed by selection with G418. Apparently, at least 59 clones have been generated. However, in this Figure several negative clones are also shown (clones 14, 33, 37 and 59). It is not clear, whether these are cells transfected with the IP3R3-Myc construct, which, for some unknown reason lost this transgene while conserving G418 resistance, or whether these are cells transfected with the empty vector, carrying the G418 resistance gene.
- Fig. 3, Panel b: Differences of the amplitude of the decrease of endoplasmic reticulum free calcium ion concentrations following carbachol treatment are shown, comparatively between wild-type and STIM1 knock-out cells. Because this is a somewhat complicated figure (“differences of differences” being shown), in the opinion of the Reviewer, it would probably be useful to include a delta sign [Δ Ca2+ ER (mM)] or something similar, on the vertical axis of the lower left graph of Panel b, and to include this Δ sign with a corresponding vertical line or arrow that spans the extent of the decrease on the two upper graphs of this Panel, where the curves turn downward following carbachol addition.
- The implications of the findings for the molecular pathomechanisms potentially involved in the etiology of Alzheimer’s disease are discussed in two separate parts, in the Introduction and the Discussion of the Manuscript. In the Introduction it is mentioned, that STIM1 has been shown to be targeted by the γ-secretase complex (lanes 67-70). On the other hand, the Authors state that Alzheimer’s disease-specific mutations in the presenilin 1/2 gene lead to the partial inhibition of presenilin/γ-secretase activity (Discussion, lane 335). Intuitively, this should lead to decreased, rather that increased proteolytic degradation of STIM1 in presenilin-mutant disease, and consequently, to increased, rather than decreased STIM1 levels. This is somewhat difficult to reconcile with the fact that in the present work Authors propose decreased STIM1 levels and its consequences on IP3R3 expression as being involved in Alzheimer’s disease. May STIM1 be involved in the etiology of Alzheimer’s disease by two, opposite mechanisms depending on presenilin mutational status? May the γ-secretase complex regulate STIM1 levels indirectly, or independently from its proteolytic activity? As Authors state in the Abstract, STIM1 decrease is observed in sporadic, as well as familial disease. The Reviewer understands that the molecular pathomechanisms of Alzheimer’s disease are not known in sufficient detail. It would be therefore very helpful for non-specialist readers to address these issues more clearly in this Manuscript.
- As shown in the Supplementary file, the expression levels of RYR2, TRPC3, NFATC4, NPR1, HTR2C and HTR1A also seem to be modified following STIM1 knock-out to extents comparable to that of ITPR3 (IP3R3). What is the position of the Authors regarding the interpretation of these raw data?
Minor:
- Lane 329: “ITPR3 expression led to a reduction of volume tumors after injection of these cells in nude mice…” : …of tumor volumes after injection… ?
- Lane 320: “CACO-2” : Caco-2; lane 328 : “DLD1” : DLD-1
- Materials and Methods : were cells monitored for mycoplasma contamination? “Foetal bovine serum” (Lane 388) means heat-inactivated fetal calf serum?
- Lane 418: was PMSF added ex tempore from a stock solution?
- Lane 427 : what was the solution used for washing the membranes?
- Lane 428: blots were incubated with “secondary antibodies (typically 1:10 000 dilution) for 1 hour at RT” : typically?
- Lane 368: The use of the SERCA2 antibody is not clear. Were SERCA levels measured in this work ?
- Lane 466: References 22, 57-60 should not be in bold.
- Lanes 501-505: Please bear in mind that dead cells may detach from the plastic surface, and this may lead to the underestimation of the extent of cell death.
- A1, bottom Panel: what is unit expression (corresponding to a fold change of 1x) ?
- Supplemental File 1: There are two spreadsheets in this file. It would be useful to include legend for both separately.
- Lane 390: Please include a Reference for collagen plating, whether collagen was dried on the plastic surface, were the dishes rinsed, source and dissolution of the collagen, etc.
- Lane 229: “Figure A4” : Figure 4.A ?. It would be also appropriate to use lower or uppercase letters homogeneously in Figures and Figure Legends.
- Lane 217: “extension” : extent ?
- Please specify in Figure Legends whether undifferentiated or differentiated SH-SY5Y cells were used. How was differentiation verified in the experiments ? Please explain briefly the advantages of using SH-SY5Y cells in the experiments (neuronal model).
- Lanes 33 and 34 : “mitochondria…” : mitochondrial… ?
- Lanes 123-125: “Remarkably, overexpression of STIM1 in SH-SY5Y cells (Figure A1, top panel) significantly increased the level of ITPR3 transcripts, without altering the level of ITPR1-2 (Figure A1, bottom panel)” : it is not clear, where are the results of STIM1 overexpression shown in Fig. 1; wild-type and STIM knock-out cells are mentioned in the Legend of Fig. 1, but STIM1 overexpression is not. Do Authors mean “normal STIM1 expression”?
- Mitochondrial calcium levels were measured using two separate probes : 4mtD3cpv (Legend, Fig 4) and mito4x-GCaMP6f (lane 214). Could the Authors discuss the reason why two distinct probes were used ? This could be done for example following Lane 487 in Materials and Methods.
- Lane 277: The meaning of the symbol (@?) preceding “slide” is not clear.
- Lanes 313 and 320 : “On the contrary…” : on the other hand ?
- STIM2 is a closely related gene to STIM1. Therefore, it would be probably of interest to the readers to discuss briefly the eventual involvement of this protein in the context of the biology of STIM1 downregulation.
